# Research on Methane Measurement and Interference Factors in Coal Mines

**DOI:** 10.3390/s22155608

**Published:** 2022-07-27

**Authors:** Xiushan Wu, Jiamin Cui, Renyuan Tong, Qing Li

**Affiliations:** 1School of Electrical Engineering, Zhejiang University of Water Resources and Electric Power, Hangzhou 310018, China; wuxs@zjweu.edu.cn (X.W.); cuijm@zjweu.edu.cn (J.C.); 2College of Mechanical and Electrical Engineering, China Jiliang University, Hangzhou 310018, China; tongrenyuan@cjlu.edu.cn

**Keywords:** methane concentration, TDLAS, environmental simulation device, fluid simulation

## Abstract

The detection of methane has always been an important part of coal mine safety. In order to improve the methane measurement accuracy in coal mines and to determine the influence of environmental interference factors on the measurement results, we designed a spherical, experimental chamber simulating the on-site environment of an underground coal mine containing methane, in which various environmental interference factors can be superimposed. The simulation chamber can generate a uniform and controllable dust environment, a controllable methane environment with concentrations below that which would trigger an alarm, controllable humidity, and environments characterized by other interference factors. Based on computational simulations of the experimental chamber with varying dust-particle-concentration distributions using a single particle size, an optimal design for the chamber has been realized in terms of the rapid mixing of dust and the flow field. Finally, we constructed an underground methane concentration measurement system for coal mines and assessed the influences of different dust concentrations and relative humidity values on the performance of methane measurements, providing a means for improving the measurement accuracy of underground coal mine, spectral, absorption-type methane sensors.

## 1. Introduction

The coal mine environment contains flammable and explosive gas, the main component of which is CH_4_. When the methane content in the environment is 5% to 16%, an explosion may occur in the case of fire, causing serious accidents [1]. Accidents caused by methane are often referred to as the “number one killer” in coal mines. According to statistics, methane accidents account for about 40% of coal mine accidents, and their associated death toll accounts for more than 30% of the total deaths in coal mine accidents [2].

In order to ensure the safety of coal mining, methane concentration monitoring in coal mines is very important. At present, methane measurement methods mainly include catalytic combustion methods [3,4,5], optical interference methods [6,7], and spectral absorption methods [8]. The instruments used in catalytic combustion methods are simple in structure and low in cost, but their measurement range is limited: most can only measure methane in the concentration range of 0–4%, the response speed is slow, the measurement accuracy is low, and it is necessary to adjust the zero-point and use a standard gas for sensitivity calibration after a period of use. Compared with catalytic combustion methods, optical interference methods have a greater measurement range and greater stability; however, the interference fringes in the measurement can only be observed manually, and the devices cannot convert the optical interference signal into a processable electrical signal. In addition, the sensors are bulky, the adjustment frequency is high, and they are easily affected by other interfering gases. Spectroscopic absorption methane sensors can mainly be divided into non-dispersive infrared (NDIR) spectroscopy [9,10] and tunable diode laser absorption spectroscopy (TDLAS), based on their working principles [11,12,13]. Infrared methane detectors based on NDIR are characterized by high cost and performance, small size, and low power consumption, but their anti-interference ability is not as good as that of TDLAS detectors [14]. A comparison of the advantages and disadvantages of methane concentration measurement methods is shown in Table 1.

Spectral absorption methane-concentration sensors have the advantages of high detection accuracy, no zero-drift, and no catalyst poisoning phenomenon (as with catalytic combustion methods). As such, they have been widely used for the measurement of methane in coal mines. Taking into consideration methane-concentration measurement using spectral absorption methods and TDLAS technology, we designed an environmental simulation device to study the methane-mixture measurement state in underground coal mines. The simulation chamber is capable of producing dust with uniform and controllable concentrations, methane gas levels with concentrations below that which would trigger an alarm, and superimposed interference factors. The device was calibrated with standard methane gas to achieve accurate methane-concentration measurements, real-time monitoring, and the wireless transmission of environmental parameters inside the simulation chamber, as well as the analysis of humidity and dust in the environment.

Based on the designed spherical, experimental chamber, the paper analyzes and studies the interference factors of the instrument for detecting methane via the TDLAS method in coal mines, providing a reference for how to correct and suppress these interference factors and improve the measurement accuracy of methane concentrations; with an integration time of 1 s, a detection sensitivity of 25 ppm is obtained, and the measurement accuracy can reach 0.06%. The paper also gives the simulation distribution of dust particles with a single particle size in the spherical, environmental simulation cabin. Through the simulation of the dust-concentration distribution and the actual experimental results, an optimization of the rapid mixing of dust in a flow field is obtained.

## 2. Design of the Spherical Simulation Chamber for Methane Measurements in Coal Mines

### 2.1. Design of Spherical Simulation Chamber for Methane Measurements

In order to simulate the air environment with a mixed state of methane and to realize methane measurements, we designed a simulation chamber structure for underground coal mine methane measurement, which is shown in Figure 1. The shape of the simulation chamber was designed to be spherical. Compared with general cylindrical and cuboid chambers, the spherical simulation chamber has no corners or edges, thus avoiding the loss of particles in the corners. Furthermore, the sphere is consistent everywhere due to its geometric structure, making the internal flow field more uniform, which can better achieve the purpose of the uniform mixing of particles. The main component of the spherical chamber is acrylic, which has good light transmittance and is convenient for observing the situation in the chamber. In order to prevent electrostatic adsorption and the loss of particles, the inner wall of the chamber is coated with a polyurethane antistatic coating, which forms a solid, crystalline, transparent coating on the acrylic surface, has good adhesion performance, and can achieve a long-term antistatic effect which is not affected by humidity. The surface resistivity is 106–109 ohms, effectively reducing the adsorption loss of static electricity to particulate matter. In the comparison of the state before and after the experiment, after the inner wall of the chamber was coated with the antistatic agent, the adsorption loss of particles was effectively reduced.

The outer diameter of the chamber is 0.5 m, and the wall thickness is 3 mm. The chamber is divided into upper and lower hemispheres, which are mounted on a ring-shaped base. The middle of the sphere can be opened. A thin rod passes through the chamber directly below the center in order to support a fan and hold a sensor. Mounted on this thin metal rod is a four-claw support plate that can slide up and down. A fan can be placed on the four-claw support plate, such that its height can be adjusted and fixed on the thin, metal rod inside the chamber.

The particulate matter airflow enters through a horn-shaped entrance on the top of the sphere. The powder outlet of the aerosol generator is connected to the large-diameter end, while the sphere is connected to the small-diameter end, such that a turbulent flow section is formed at the entrance. Then, after passing through the turbulent flow section, the dust is gradually and uniformly diffused throughout the sphere [15].

### 2.2. Working Principle of the Simulation Chamber

The designed underground coal mine, methane simulation chamber includes a gas-quality controller, an aerosol generator, and a parameter measurement and control system. The parameter measurement and control system is mainly composed of temperature, humidity, and air-pressure sensors; laser dust-sensors; a TDLAS methane-concentration measurement module; a data acquisition module; a wireless transceiver module; and a display module. The methane-concentration measurement and control system is depicted in Figure 2 [16].

The two gas-quality control modules of the system respectively adjust the methane concentration and humidity in the spherical chamber by adjusting the intake air flow. The methane gas introduced into the simulation chamber is provided by 99.9% methane standard gas cylinders. Strict calculation and control ensure that the methane concentration in the chamber is always far below the methane explosion lower limit of 5.0%, such that the final methane gas concentration in the simulated chamber is in the range of 0–1.0%. Before the methane gas is introduced, the oxygen in the chamber needs to be completely discharged, and the experimental procedure should be carried out in a well-ventilated place. In order to study the effect of humidity on methane measurements, high-humidity nitrogen is added to the chamber after pure water, as nitrogen is used as the dilution gas without affecting the control over the methane environment. The system can realize the adjustment of humidity with an adjustment range of 20–90%.

The parameter measurement control system is mainly composed of temperature and humidity sensors, air-pressure sensors, dust-concentration sensors, and the measurement control circuit. The temperature, humidity, and air-pressure sensors in the chamber realize the measurement of temperature, humidity, and air pressure, respectively, while the dust-concentration sensor realizes the function of the real-time measurement of dust concentrations in the chamber. Meanwhile, the TDLAS methane-concentration measurement module realizes methane-concentration measurements in the chamber and transmits the obtained data to the upper computer. The principle of sensor selection was that they must be small in size and high in precision in order to reduce their influence on the flow field in the chamber. During installation, the temperature and humidity sensors were placed on the inner wall of the compartment, and the sensor pins led out via the hole on the surface of the compartment, with wires connected to the measurement control system. The air-pressure, dust-concentration, and methane-concentration sensors were attached to the outer wall of the dome and conducted measurements through sampling holes on the sphere’s outer surface.

The used temperature and humidity sensor chip is a digital temperature and humidity sensor SHT20 (Sensirion, Stafa, Switzerland). The air-pressure sensor is an XGZP6847, with a relative pressure measurement range of 0–200 kPa, a resolution of 2 kPa, and a working temperature range of −20–100 °C. The dust-concentration sensor is an SDS198 (Nuofang, Jinan, China), which uses the principle of laser scattering to obtain the mass concentration of suspended particles in the air, ranging from 1 to 100 microns. The data are stable and reliable, are output digitally, and have a high degree of integration. The range is 0–20 mg/m^3^, the relative error has a maximum value of ±20% and ±30 μg/m^3^, and the size of the sensor is only 71 × 70 × 23 mm^3^.

The dust particles inside the chamber are provided by an aerosol-generating device, which controls the concentration, particle size, and flow rate of the particles introduced into the chamber. For this purpose, we adopted a vibrating hole-type monodisperse aerosol generator, FMAG 1520 (MSP, St. Paul, MN, USA), and a condensed core-type monodisperse aerosol generator, MAG3000 (PALAS, Karlsruhe, Germany) [17].

## 3. Methane Detection Based on TDLAS

### 3.1. Analysis of the Harmonic Detection Principle

TDLAS technology tunes the output wavelength of a semiconductor laser using the current and temperature, scans a certain absorption line of the measured substance, and obtains the concentration of the measured substance by detecting the absorption intensity of the absorption spectrum. TDLAS detects the number of molecules that the laser passes through in the gas channel under test, and the obtained gas concentration is the average concentration of the entire channel. The quantitative gas concentration calculation in TDLAS is based on the Beer-Lambert law, which describes the relationship between light absorption and light passing through the detected substance. The change in light intensity of a beam passing through a test gas is given by:(1)$$I_{t}=I_{0}\exp[-PS(T)\varphi (v)CL]$$
where *I*_0_ is the laser output intensity, *I_t_* is the laser detection light intensity after gas absorption, *P* is the gas pressure, *S*(*T*) (cm^−2^ atm^−1^) is the characteristic spectral-line intensity of the gas, *φ*(*v*) is a linear function, *C* is the concentration of the measured gas, and *L* is the optical path.

Harmonic detection is one of the most effective methods for extracting small signals from strong interference noise and improving detection sensitivity. Using a tunable light source with harmonic absorption as a method to detect the concentration of methane gas can effectively avoid the interference of environmental factors. It is assumed that low-frequency triangular and high-frequency sine waves are injected into the DC drive current of the distributed laser in order to realize laser-wavelength scanning and modulation. The frequency of the modulated laser can be expressed as *v* = *v*_0_ + *v_f_*sin(*ωt*), where *v*_0_ is the center frequency of the light source when it is not modulated, and *v_f_* is the frequency modulation amplitude. When the absorption is weak (i.e., *PS*(*T*) *φ*(*v*)*LC* << 1), we define *σ*(*v*) as the absorption rate of the gas to be measured at frequency *v*. Then, Formula (1) can be simplified as:(2)$$I_{t}\approx I_{0}[1-PS(T)\varphi (v)CL]=I_{0}[1-\sigma (v)L]$$

Expanding *σ*(*v*) into a Fourier series gives:(3)$$\sigma (v_{0}+v_{f}\cos\omega t)=\sum_{n=0}^{\infty }H_{n}(v_{0})\cos n\omega t$$
where *H_n_*(*v*_0_) represents the Fourier coefficient of the *n*th harmonic component. The detector signal is input into a lock-in amplifier, and each harmonic component proportional to *I_n_H_n_*(*v*_0_)*L* can be obtained. Under normal temperature and pressure, the Lorentz function is generally used to describe the absorption line of a gas, namely:(4)$$\sigma (v)=\frac{1}{1+{\left(\frac{v-v_{c}}{\Delta v}\right)}^{2}}$$
where *v_c_* is the center frequency of the gas absorption peak, and ∆*v* is the full width at half-maximum of the gas absorption line. When the center frequency of the light source is modulated to the gas absorption peak (i.e., *v*_0_ = *v_c_*), a modulation coefficient *m* = *v_f_/*∆*v* is defined. According to the expression of the second harmonic given by Reid et al., the second harmonic at the center of the absorption line can be obtained. The signal peak expression is:(5)$$P_{2f}\propto \frac{I_{0}S(T)PCL}{\pi \Delta v_{c}}\{\frac{2}{m^{2}}\left[\frac{2+m^{2}}{{\left(1+m^{2}\right)}^{1/2}}-2\right]\}$$

From Equation (5), it can be concluded that the peak value of the second harmonic is proportional to the gas concentration, in which only the gas concentration is unknown. As long as a standard gas is used for calibration, the concentration inversion can be performed using the relationship between the gas concentration and the second harmonic peak. Thus, laser-modulation absorption-spectrum-detection technology can effectively overcome the low-frequency noise in the circuit and the optical path, consequently improving the measurement sensitivity. Notably, this is the most commonly used method in TDLAS technology.

### 3.2. Design of TDLAS Methane Detection Device

Methane-gas-concentration measurement methods based on TDLAS technology can be divided into two types: direct absorption methods and modulation methods. When using a direct absorption method to measure the gas concentration, as the gas absorption is very weak, the light signal received by the detector is also very weak, and so, it is easily affected by environmental noise and stray light, resulting in poor accuracy and sensitivity. Generally, this technology is used in the presence of large concentrations or long optical-path measurements. Modulation technology is generally divided into two types: frequency modulation (FMS) and wavelength modulation. Due to the difficulties associated with frequency modulation technology, wavelength modulation technology is more commonly used. The designed methane-gas-concentration measurement system based on TDLAS technology is shown in Figure 3, which consists of modules including a laser, laser controller, photodetector, lock-in amplifier, and data collector [18].

The laser in Figure 3 uses a 1653 nm, tunable, distributed feedback laser as the light source [19]. Its linewidth is generally within 1 MHz, which is much smaller than the traditional infrared light source, and the frequency of the laser can be changed within a small range. This is convenient for selecting the absorption peak of the gas to be tested and reducing the interference of other background gases. The laser controller is used for temperature adjustment and current modulation in order to control the wavelength emitted by the laser at the corresponding absorption peak of methane gas. The distributed feedback laser uses a 5 kHz sine-wave modulation signal and a 20 Hz triangle-wave signal to modulate the laser wavelength [20]. The 5 kHz modulation signal is provided by the laser controller, while the 20 Hz scan signal is provided by the lock-in amplifier. The laser controller uses the Thorlabs ITC4005, which can provide a precise and stable current to the laser diode, and it has excellent temperature stability, with an offset of only 0.002 degrees Celsius over 24 h. The photodetector collects the outgoing light, converts the light signal into a voltage signal, and transmits it to the lock-in amplifier. The lock-in amplifier first provides a scanning signal to the laser. Next, the outgoing light-intensity signal received by the detector is extracted by the lock-in amplification technology in order to extract the second harmonic signal and then send it to the data collector to detect the concentration of methane gas. The lock-in amplifier of this device is an SR860 (SRS, Sunnyvale, CA, USA), which has a bandwidth of 500 kHz, low-noise voltage and current input, and high-bandwidth output, and it can provide a 20 Hz scanning signal to the laser. The data acquisition module is specifically used to collect the second harmonic peak signal output by the lock-in amplifier in the TDLAS methane-measurement system and transmit it to the host computer, while the host computer’s software converts the second harmonic peak value into methane-concentration information and displays it [21]. The side view of the constructed experimental measurement platform and the used measuring instruments is shown in Figure 4a, the top view of the experimental measurement platform is shown in Figure 4b, and the actual setup of the entire system is shown in Figure 4c.

## 4. Measurement Results

The second harmonic detection peak-to-peak value measured for standard methane gas can be used invert the expression between the second harmonic detection peak-to-peak value and the methane concentration. The measured second harmonic detection peak-to-peak relationship, with respect to the methane concentration, is provided in Table 2. Assuming that the methane concentration is the independent variable *x,* and the peak-to-peak value of the second harmonic is the dependent variable *y,* the relationship is shown in Figure 5. The relationship between the peak-to-peak value of the second harmonic detection and the methane concentration was obtained as *y* = 47.94 + 91.93*x*, showing a linear correlation; with this, methane-concentration detection can be realized [22,23].

## 5. Influence of Environmental Parameter Changes on Measurement Results and Experiments

### 5.1. The Influence of Dust Concentrations on Methane Measurements

The dust concentration range in an underground coal mine at the working face can reach up to 1000 mg/m^3^, while the maximum can be 100 mg/m^3^ in the roadway. Therefore, the influence of dust concentrations on methane measurements must be considered. In order to simulate the influence of different concentrations of dust on the methane-measurement results, a trumpet-shaped opening was included on the top of the designed simulation chamber, and the powder outlet of the aerosol generator was connected to the large-diameter end, while the inlet of the chamber was connected to the small-diameter end. A turbulent flow section is thus formed at the top of the sphere, where the airflow with particulate matter enters the chamber. This turbulent flow allows the dust to diffuse into the sphere gradually and uniformly.

In order to produce uniform dust at certain concentrations, the placement, number, cross-sectional area, wind speed, and flow rate of the fans inside the chamber needed to be set according to the uniformity of the mixing of the particles. Complete mixing was required to obtain effective experimental results. First, Fluent software was used to simulate the distribution of particle concentrations in different areas in order to select the best incoming particle flow, fan placement, number, cross-sectional area, wind speed, and other parameters.

In the simulation, the wind speed should not be too high. If it is too high, the particles will move violently, and collision loss will occur. According to the actual experimental situation, the pressure difference of the fan was generally less than 0.3 Pa; the position of the fan was symmetrical or at the central axis to generate a uniform flow field; and the cross-sectional area of the fan (i.e., the size) and number of fans were selected according to the volume of the experimental chamber. The fewer the better, as more fans will affect the dispersion of the particles according to the volume of the chamber and the selected fans. Therefore, in this paper, we only considered configurations of up to four fans.

The experimental conditions set during the simulation were as follows: The diameter of the sphere was 500 mm; the air inlet and outlet were situated at the top and bottom of the sphere, respectively; the diameter of the inlet/outlet was 10 mm; the air intake flow rate was 3 L/min; the air intake mixed particulate matter was comprised of NaCl particles; the air intake diameter was 2.5 mm; and the mass concentration of the particulate matter was 1 mg/m^3^.

Figure 6a shows the simulation results when considering free diffusion at the inlet without a fan. A fan was then simulated with a diameter of 80 mm and placed in the center of the chamber. The simulation results under a pressure of 0.1 Pa in front of and behind the fan are shown in Figure 6b. Then, four fans were placed in the upper and lower hemispheres of the sphere, with two fans in each hemisphere. Figure 6c shows the simulation result with the 4 fans under a pressure of 0.1 Pa in front of and behind the fans, while Figure 6d shows the result when using a single fan with a pressure of 0.5 Pa. In Figure 6, the picture on the left shoes the mass-concentration simulation result, and the right shows wind-speed simulation results.

In the simulation diagrams shown in Figure 6, the redder areas indicate higher particle concentrations, especially closer to the inlet. When there was no fan, the dust concentration was concentrated on the central axis. When there was a single fan at the central axis, the dust concentration distribution around the spherical wall was very uniform although some areas near the center of the sphere presented lower concentrations. In the simulation with four fans, the dust concentration in the central area was significantly higher than that generated using only one fan, but the distribution of the dust concentration on the spherical wall was not as uniform. Considering these results, we chose to place 1 fan on the central axis of the experimental chamber, and the simulation effect was best when the pressure difference was 0.1 Pa. Figure 7 depicts the flow field when using 1 fan on the central axis, which allowed the dust to be evenly distributed throughout the chamber.

In the actual experiment, the PM2.5 particles emitted by the vibrating, hole-type, aerosol generator were passed into the spherical chamber for mixing according to the simulation results, and the dust concentration was detected at each sampling point using a laser dust-detector. There were eight sampling points in the upper hemisphere and eight in the lower hemisphere. Every group of four sampling points lay in a plane: in the upper hemisphere, points 1–4 and 5–8 were coplanar, while in the lower hemisphere, points 9–12 and 13–16 were coplanar. A schematic diagram of the sampling points is shown in Figure 8. Table 3 provides the experimental measurement results of the dust-mass concentration at each sampling point in the chamber for 3 time points [24].

It can be seen from Table 3 that the standard deviation between the 4 sampling points on each plane was very small due to the symmetry of the sphere. It should be noted that the concentrations at the sampling points in the lower hemisphere were slightly higher than those in the upper hemisphere, which is consistent with the weight of the particles and the corresponding sedimentation phenomena. Overall, the particles are mixed evenly and, from the perspective of the time axis, the concentration at each sampling point after 1 h was relatively stable.

In addition to the error between each sampling point, the repeatability error of each sampling point under the same experimental conditions (i.e., with constant particle concentration, flow rate, and introduction time) to reach a stable concentration state is also very important. The repeatability error calculation formula is:(6)$$RSD=\sigma /\bar{X}$$
where *RSD* is the repeatability error, *σ* is the standard deviation of each experiment at each sampling point, and $\bar{X}$ is the average value of six experiments at each sampling point.

For the situation in which four sampling points on different horizontal planes were selected to conduct six experiments under the same conditions, the experimental measurement results are shown in Table 4. From the table, it can be concluded that the repeatability error at each sampling point did not exceed 6%, indicating that the experimental scheme and design were correct, such that the experiment is reproducible.

Next, we set different concentrations of particles in the simulation chamber and conducted measurements on standard methane gas. The measurement results are shown in Table 5. When the concentration of dust was low, the impact on the methane sensor was almost negligible; however, when the dust concentration reached a certain level, it caused a large error. In view of the high dust-concentration at the working face in an underground mine, it is necessary to consider the impact of high dust concentrations on the results when measuring, and appropriate measurement correction needs to be carried out.

### 5.2. Effect of Humidity on Methane Concentration Measurements

The actual methane concentration measurement should be performed in an air atmosphere. However, in order to obtain the influence of varying environmental humidity levels on methane measurements, high-humidity N_2_ was added to the chamber after pure water in order to change the humidity of the simulated chamber, which was then passed into the TDLAS methane detection device. This method not only achieves humidification in the cabin, but the N_2_ also acts as a diluent for methane, thus requiring no additional steps for the control of the methane-concentration environment.

The relationship between the second harmonic and the relative humidity of the environment is shown in Table 5. From the measurement results in Table 6, it can be concluded that the relative humidity in the environment did not affect the peak value of the second harmonic; however, relative humidity will affect the light intensity received by a photodetector.

## 6. Conclusions

Based on the background of methane detection in coal mines, in this paper, we focused on analyzing the mechanism and interference factors of methane measurement using a spectral absorption method, and we designed a methane measurement device and a spherical experimental chamber for simulating the underground environment of coal mines. The following conclusions were drawn, based on theoretical analysis and experimental results.

Based on TDLAS detection technology, combined with a tunable light source and harmonic detection method, a methane detection device was built, which was installed in the spherical environment simulation chamber.By measuring the methane concentrations at different dust concentrations, it can be concluded that the dust concentration has a great influence on the measurement of methane concentrations. When the dust-mass concentration is 1.5–1.9 mg/m^3^, the relative measurement error can reach about 6%; when it is 10 ± 10% mg/m^3^, the relative measurement error can reach about 10%; and when it is 20 ± 10% mg/m^3^, the relative error can reach about 16%. Therefore, methane-concentration measurement results should be corrected at high dust concentrations.Based on the designed spherical, environment-simulation chamber, the experiments on the effects of humidity on the measurement of methane concentration were completed. Experimental results show that, when the standard methane concentration is 0.20%, the maximum measurement error is less than 4%, and when the standard concentration is 1.0%, the relative error of the measurement can be ignored.In this paper, through theoretical analysis and simulation leading to the construction of an experimental device, we finally verified methane-measurement performance experimentally, thus providing a reference for improving the measurement accuracy of spectral absorption methane sensors in coal mines.

Our future work will focus on the study of the influence mechanism of particle size, shape, scattering angle, and such on light-scattering measurement results, thereby improving measurement accuracy. Second, the lock-in amplifier is the key to the TDLAS measurement of methane; its performance directly affects the detection results. Future research will study the miniaturization of the lock-in amplifier and work towards designing a high-performance lock-in amplifier module, so as to further improve the detection precision of the instrument.

## Figures and Tables

**Figure 1 sensors-22-05608-f001:**
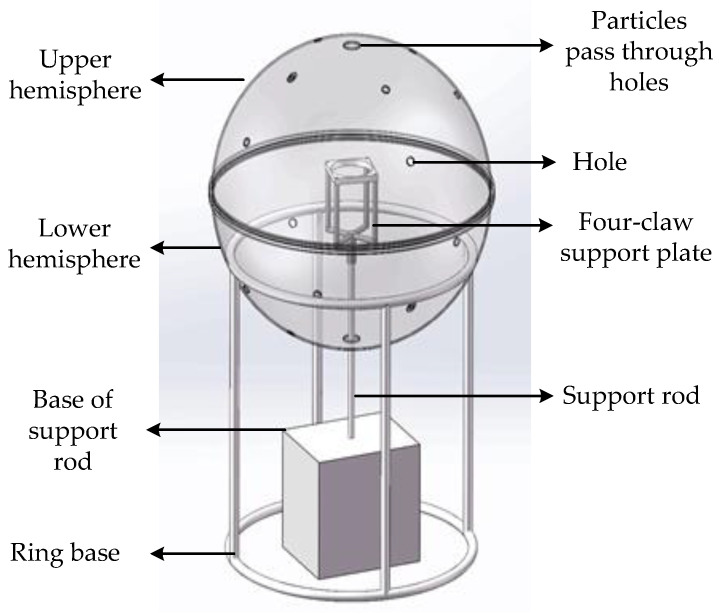
Schematic diagram of the methane-measurement simulation chamber.

**Figure 2 sensors-22-05608-f002:**
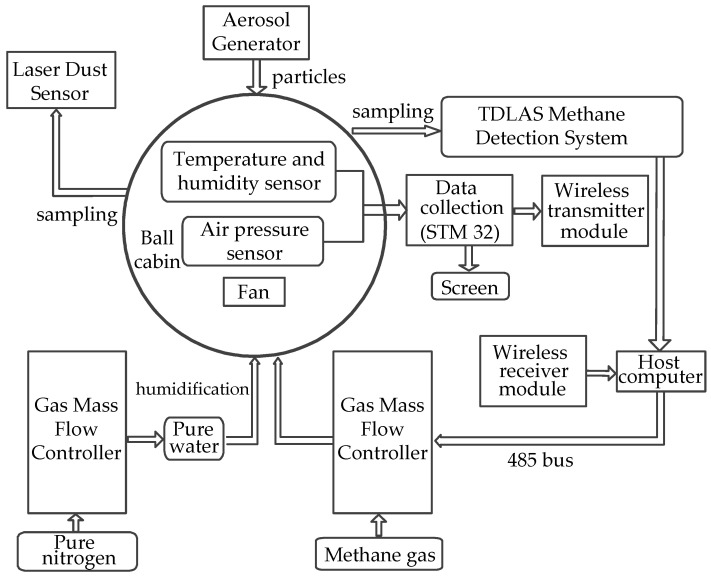
Block diagram of the methane-concentration measurement and control system.

**Figure 3 sensors-22-05608-f003:**
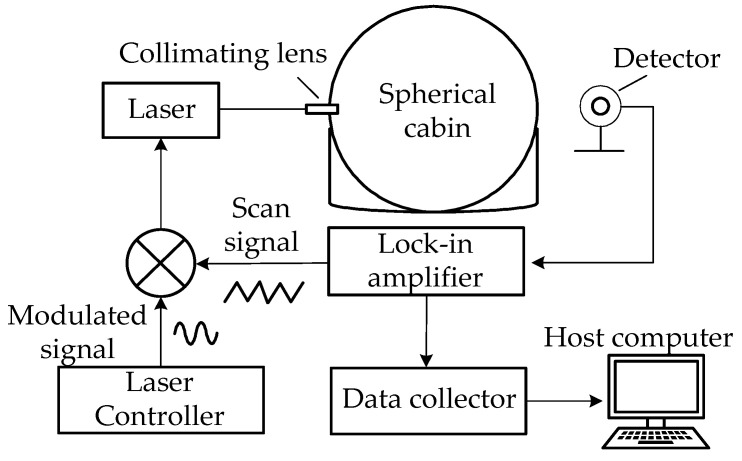
Methane measurement system based on TDLAS.

**Figure 4 sensors-22-05608-f004:**
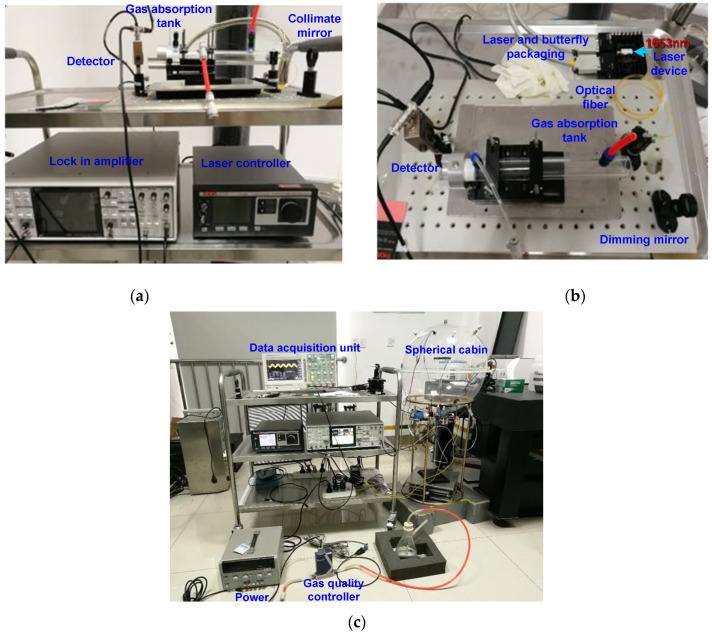
Measuring system: (**a**) Measuring instruments and platform; (**b**) measurement platform; (**c**) the actual setup of the entire system.

**Figure 5 sensors-22-05608-f005:**
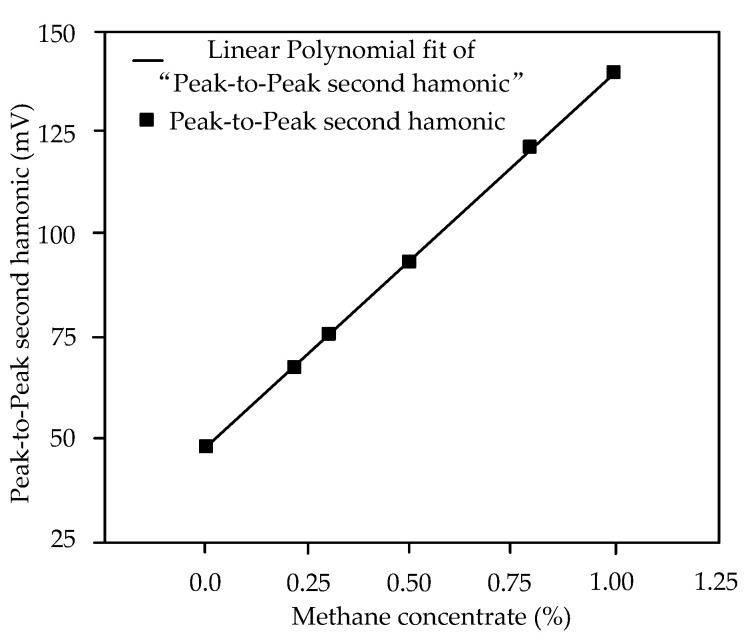
Relationship between peak-to-peak average values of second harmonic detection and methane.

**Figure 6 sensors-22-05608-f006:**
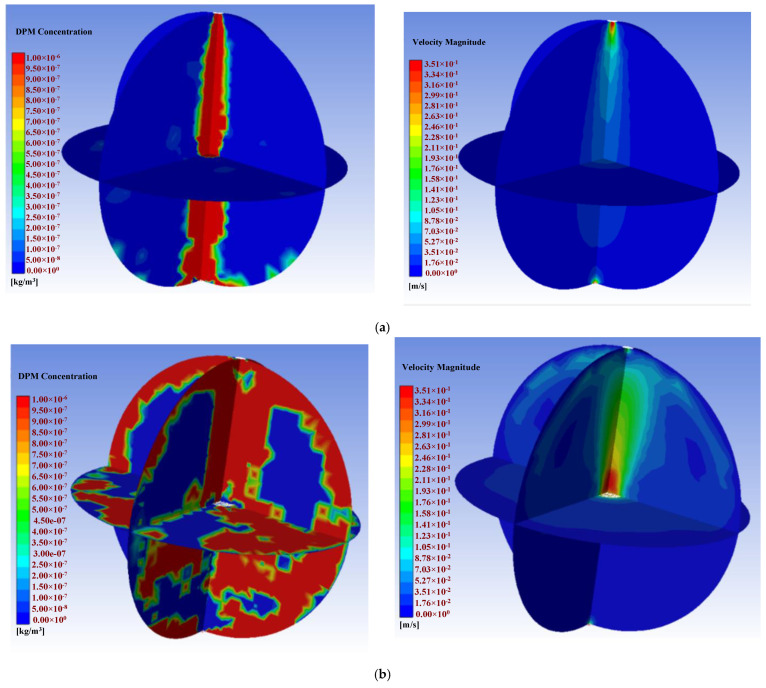
Particle-concentration and wind-speed simulation results under different wind fields: (**a**) No fan; (**b**) 1 fan with a pressure of 0.1 Pa in front of and behind the fan; (**c**) 4 fans with a pressure of 0.1 Pa; (**d**) 1 fan with a pressure of 0.5 Pa.

**Figure 7 sensors-22-05608-f007:**
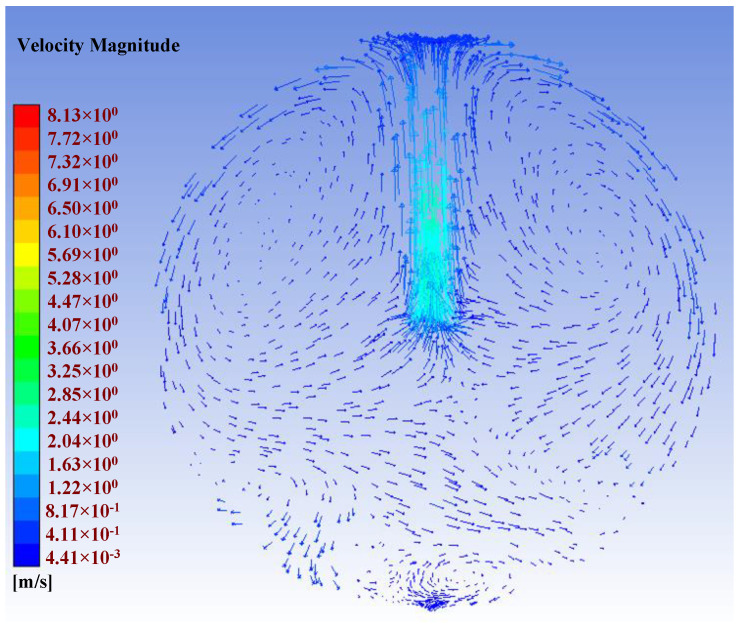
Flow field diagram when a single fan was placed on the central axis.

**Figure 8 sensors-22-05608-f008:**
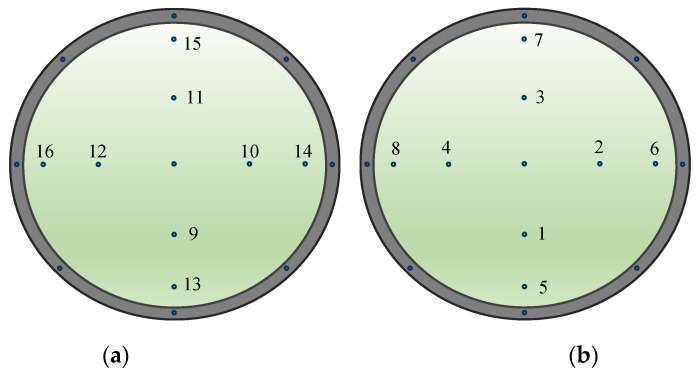
Schematic diagram of sampling-point numbering in the spherical chamber: (**a**) Bottom view of sampling points in the lower hemisphere; (**b**) top view of the sampling points in the upper hemisphere.

**Table 1 sensors-22-05608-t001:** Comparison of methane concentration measurement methods.

Measurement Methods	Advantages	Disadvantages
Catalytic combustion	Simple structure and low cost.	Conductivity sensor is obviously affected by the processing. Large measurement error for low concentrations of methane. The measurement result is easily affected by air humidity and oxygen, and it is difficult to overcome with zero drift.
Optical Interferometry	Larger measurement range and stronger stability compared with catalytic combustion.	The interference fringes can only be observed by the naked eye, and the optical interference signal cannot be converted into a processable electrical signal. The sensor is large in size, has a high adjustment frequency, and is greatly affected by other interfering gases.
Infrared spectroscopy of NDIR	Cost-effective, small size, and low power consumption.	Strong anti-interference ability, but not as good as TDLAS.
Infrared spectroscopy of TDLAS	Real-time online. Fast and high sensitivity. Little impact on the environment. Better wavelength selectivity.	Expensive and high technical requirements.

**Table 2 sensors-22-05608-t002:** Experimental results of the relationship between the second harmonic peak-to-peak value and the methane concentration.

Methane Concentration (%)	Measured Peak-to-Peak Value of the Second Harmonic (mV)		
	Minimum	Maximum	Average
0.00	44.0	52.8	48.4
0.22	63.2	72.0	67.6
0.30	70.4	78.5	75.5
0.50	89.9	96.6	93.9
0.80	116.8	124.2	121.5
1.00	137.0	143.0	140.0

**Table 3 sensors-22-05608-t003:** Dust-mass-concentration-distribution measurement results.

Sampling Point	Dust-Mass-Concentration Measurement Results (mg/m^3^)		
	59th Second	119th Second	239th Second
1	0.099	0.101	0.103
2	0.101	0.098	0.098
3	0.104	0.106	0.107
4	0.103	0.104	0.099
5	0.095	0.096	0.093
6	0.090	0.093	0.091
7	0.095	0.098	0.096
8	0.092	0.091	0.094
9	0.114	0.113	0.117
10	0.118	0.117	0.119
11	0.121	0.125	0.117
12	0.122	0.123	0.116
13	0.100	0.102	0.102
14	0.112	0.114	0.111
15	0.112	0.111	0.113
16	0.111	0.116	0.118

**Table 4 sensors-22-05608-t004:** Repeatability test results of dust concentration at four sampling points.

Experiment Number and Parameters	Sample Point 1 (mg/m^3^)	Sample Point 5 (mg/m^3^)	Sample Point 9 (mg/m^3^)	Sample Point 13 (mg/m^3^)
1	0.099	0.095	0.114	0.100
2	0.085	0.097	0.111	0.102
3	0.091	0.095	0.099	0.114
4	0.089	0.090	0.098	0.111
5	0.095	0.093	0.103	0.112
6	0.090	0.083	0.101	0.113
$\bar{X}$	0.0915	0.0922	0.1043	0.1087
σ	0.0045	0.0046	0.0061	0.0055
*RSD*	4.88%	5.03%	5.80%	5.09%

**Table 5 sensors-22-05608-t005:** Influence of dust-mass concentration on the TDLAS methane sensor.

Methane Standard Gas Concentration (%)	Average Value of Measurements at Different Dust-Mass Concentrations (%)					
	Dust-Mass Concentration 1.5–1.9 mg/m^3^		Dust-Mass Concentration 10 ± 10% mg/m^3^		Dust-Mass Concentration 20 ± 10% mg/m^3^	
	Measurement Value (%)	Relative Error	Measurement Value (%)	Relative Error	Measurement Value (%)	Relative Error
0.20	0.19	5.00%	0.18	10.00%	0.17	15.00%
0.30	0.28	6.66%	0.27	10.00%	0.25	16.70%
0.50	0.47	6.00%	0.45	10.00%	0.42	16.00%
0.80	0.76	5.00%	0.75	6.25%	0.70	12.50%
1.00	0.96	4.00%	0.94	6.00%	0.90	10.00%

**Table 6 sensors-22-05608-t006:** Effect of humidity on methane measurement.

Relative Humidity (%)	Parameters	Methane Standard Gas Concentration (%)				
		0.20	0.30	0.50	0.80	1.00
20	Second harmonic peak-to-peak (mV)	67.0	74.8	94.3	121.0	140.4
	Inverse calculation of methane concentration (%)	0.207	0.292	0.504	0.795	1.006
	Relative error (%)	3.67	2.61	0.86	0.66	0.58
40	Second harmonic peak-to-peak (mV)	66.9	74.9	94.2	121.1	140.3
	Inverse calculation of methane concentration (%)	0.206	0.293	0.503	0.796	1.005
	Relative error (%)	3.12	2.24	0.64	0.52	0.47
60	Second harmonic peak-to-peak (mV)	66.9	75.1	94.6	121.4	139.9
	Inverse calculation of methane concentration (%)	0.206	0.295	0.508	0.799	1.000
	Relative error (%)	3.12	1.52	1.51	0.11	0.03
80	Second harmonic peak-to-peak (mV)	66.9	74.8	94.6	121.2	140.0
	Inverse calculation of methane concentration (%)	0.206	0.292	0.508	0.797	1.001
	Relative error (%)	3.12	2.61	1.51	0.39	0.14

## Data Availability

Not applicable.

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
