# Peer review of "Research on Methane Measurement and Interference Factors in Coal Mines"

_sensors, 2022, doi:10.3390/s22155608_

Round 1

Reviewer 1 Report

Reviewer’s Comments

The authors presented the influence of small particles on the laser-type methane sensor. However, I have several unclear points in the manuscript. Please make clear the followings.

1.        Figure 4; you explained Fig. 4 (a) is the entire setup, but the figure doesn’t contain the parts of Fig. 4 (a) and Fig. 4 (b).

2.        Figure 6; the size of the text in the figures is too small to read.

3.        The text after Fig. 6 explained the simulation results of particle distribution in the box. However, I couldn’t catch the information on how these distributions influence methane detection.

4.        Page 12, line 1-2; you wrote “High-humidity N2 was added to the chamber after pure water, in order to change the humidity of the simulated chamber, which was then passed into the TDLAS methane detection device.”. I think that experiments should be examined in the air atmosphere.

Author Response

Thank you very much for your letter and for the reviewers’ comments concerning our manuscript entitled “Research on Methane Measurement and Interference Factors in Coal Mines” (ID: sensors-1790410). Those comments are all valuable and very helpful for revising and improving our paper, as well as the important guiding significance to our researches. We have studied comments carefully and have made correction which we hope to meet with approval. Revised portions are marked in yellow in the paper. The main corrections in the paper and the responds to the reviewer’s comments are as flowing:

The authors presented the influence of small particles on the laser-type methane sensor. However, I have several unclear points in the manuscript. Please make clear the followings.

  1. Figure 4; you explained Fig. 4 (a) is the entire setup, but the figure doesn’t contain the parts of Fig. 4 (a) and Fig. 4 (b).

Response: According to your suggestions,the explanation of Figure 4 has been modified, and the specific explanation is added in lines 216-219 of the article, and highlighted in yellow.

  1. Figure 6; the size of the text in the figures is too small to read.

Response: According to your suggestions,Figure 6 and Figure 7 have been modified, and the numbers shown in the figures are bolded. Please refer to Figures 6 and 7 for details. 

  1. The text after Fig. 6 explained the simulation results of particle distribution in the box. However, I couldn’t catch the information on how these distributions influence methane detection.

Response: The influence of dust concentration on the detection accuracy of methane is shown in the measurement data in Table 4.

  1. Page 12, line 1-2; you wrote “High-humidity N2 was added to the chamber after pure water, in order to change the humidity of the simulated chamber, which was then passed into the TDLAS methane detection device.”. I think that experiments should be examined in the air atmosphere.

Response: Your suggestion is correct, the actual methane concentration measurement should be done in the air atmosphere. However, in order to obtain the influence of various environmental humidity on methane measurement, this paper designs a simulated experimental cabin to obtain the measurement of methane concentration under different humidity conditions.

Thank you again for your comments, and we hope our responses and corrections would meet with your approval.

Reviewer 2 Report

I consider the proposed device for simulating various scenarios of methane concentration in conditions similar to mine conditions as an interesting idea. However, the authors did not emphasize enough technical advantages over standard methane measurements in a mine. Below are my other comments:

page 2, I think it would be worth compiling the advantages and disadvantages of individual measuring devices in a table for better readability of the article.

page 2, what is the scientific novelty of this work? please specify it.

page 5, the authors wrote that the absorption spectrum with laser modulation improves the sensitivity of the measurement. As far as this sensitivity is improved (please provide information as a percentage)

page 11, what must the RSD value be to say that the results are repeatable?

Table 4, what is the limit value of dust concentration for a correct methane measurement?

page 13, there is no comparison of the results obtained from the tests with the results obtained with the devices mentioned at the beginning of the article. What did the proposed measuring system bring? What is the added value, please write it down better in the conclusions.

Author Response

Thank you very much for your letter and for the reviewers’ comments concerning our manuscript entitled “Research on Methane Measurement and Interference Factors in Coal Mines” (ID: sensors-1790410). Those comments are all valuable and very helpful for revising and improving our paper, as well as the important guiding significance to our researches. We have studied comments carefully and have made correction which we hope to meet with approval. Revised portions are marked in yellow in the paper. The main corrections in the paper and the responds to the reviewer’s comments are as flowing:

I consider the proposed device for simulating various scenarios of methane concentration in conditions similar to mine conditions as an interesting idea. However, the authors did not emphasize enough technical advantages over standard methane measurements in a mine. Below are my other comments:

page 2, I think it would be worth compiling the advantages and disadvantages of individual measuring devices in a table for better readability of the article.

Response: According to your suggestions,a comparison of the advantages and disadvantages of methane concentration measurement methods is shown in Table 1.  The Table 1 is added in the page 2.

page 2, what is the scientific novelty of this work? please specify it.

Response: According to your suggestions, the scientific novelty of this work is added in the page 2 of  line 65-71.

page 5, the authors wrote that the absorption spectrum with laser modulation improves the sensitivity of the measurement. As far as this sensitivity is improved (please provide information as a percentage)

Response: According to your suggestions, relevant data has been added in line 65-69.

page 11, what must the RSD value be to say that the results are repeatable?

Response: Repeatability error refers to the random error obtained by the same operator performing multiple consecutive measurements of the same input value from the same direction, and within the full measurement range and under the same working conditions. In industrial applications, this value is generally lower than 5% can be to say that the results are repeatable.

Table 4, what is the limit value of dust concentration for a correct methane measurement?

Response: In the experiment, see Table 5,the dust concentration has a great influence on the measurement of methane concentration. To determine the dust concentration according to the actual application, the measurement data must be corrected.

page 13, there is no comparison of the results obtained from the tests with the results obtained with the devices mentioned at the beginning of the article. What did the proposed measuring system bring? What is the added value, please write it down better in the conclusions.

Response: According to your suggestions, the conclusions of the article have been rewritten in the line 358-376.

Thank you again for your comments, and we hope our responses and corrections would meet with your approval.

Round 2

Reviewer 1 Report

Reviewer’s Comments

I’m not a specialist in this kind of calculation, but a researcher of gas sensors. Please explain your results more carefully.

  1. Figure 4; you explained Fig. 4 (a) is the entire setup, but the figure doesn’t contain the parts of Fig. 4 (a) and Fig. 4 (b).

Response: According to your suggestions,the explanation of Figure 4 has been modified, and the specific explanation is added in lines 216-219 of the article, and highlighted in yellow.

Question: You wrote “The constructed experimental measurement platform and used is shown in Figure 4 (a), the experimental measurement platform is shown in Figure 4 (b),”. You just wrote the same sentence twice. I think the upper part of Fig. 4 (a) and Fig. 4 (b) should be the same exactly. You changed the setup a little bit. It is confusing to understand Fig. 4 (a) and (b). Why don’t you take photos again?

  1. Figure 6; the size of the text in the figures is too small to read.

Response: According to your suggestions,Figure 6 and Figure 7 have been modified, and the numbers shown in the figures are bolded. Please refer to Figures 6 and 7 for details.

 Comment: The size of the characters is too small to read. Could you enlarge more?

  1. The text after Fig. 6 explained the simulation results of particle distribution in the box. However, I couldn’t catch the information on how these distributions influence methane detection.

Response: The influence of dust concentration on the detection accuracy of methane is shown in the measurement data in Table 4.

Question: Please add your explanation to the manuscript. Table 4 seems to be the list of dust concentrations.

  1. Page 12, line 1-2; you wrote “High-humidity N2 was added to the chamber after pure water, in order to change the humidity of the simulated chamber, which was then passed into the TDLAS methane detection device.”. I think that experiments should be examined in the air atmosphere.

Response: Your suggestion is correct, the actual methane concentration measurement should be done in the air atmosphere. However, in order to obtain the influence of various environmental humidity on methane measurement, this paper designs a simulated experimental cabin to obtain the measurement of methane concentration under different humidity conditions.

Question: Your answer should be contained in the manuscript.

Author Response

Thank you very much for your letter and for the reviewers’ comments concerning our manuscript entitled “Research on Methane Measurement and Interference Factors in Coal Mines” (ID: sensors-1790410). Those comments are all valuable and very helpful for revising and improving our paper, as well as the important guiding significance to our researches. We have studied comments carefully and have made correction which we hope to meet with approval. Revised portions are marked in yellow in the paper. The main corrections in the paper and the responds to the reviewer’s comments are as flowing:

The authors presented the influence of small particles on the laser-type methane sensor. However, I have several unclear points in the manuscript. Please make clear the followings.

  1. Figure 4; you explained Fig. 4 (a) is the entire setup, but the figure doesn’t contain the parts of Fig. 4 (a) and Fig. 4 (b).

Response: According to your suggestions,the explanation of Figure 4 has been modified, and the specific explanation is added in lines 216-219 of the article, and highlighted in yellow.

Question: You wrote “The constructed experimental measurement platform and used is shown in Figure 4 (a), the experimental measurement platform is shown in Figure 4 (b),”. You just wrote the same sentence twice. I think the upper part of Fig. 4 (a) and Fig. 4 (b) should be the same exactly. You changed the setup a little bit. It is confusing to understand Fig. 4 (a) and (b). Why don’t you take photos again?

Response: According to your suggestions,the explanation of Figure 4(a) and 4(b) have been modified in the lines 223-226. What the author wants to express is that the  Figure 4(a)  is the side view,   Figure 4(b) is the top view.

  1. Figure 6; the size of the text in the figures is too small to read.

Response: According to your suggestions,Figure 6 and Figure 7 have been modified, and the numbers shown in the figures are bolded. Please refer to Figures 6 and 7 for details.

Comment: The size of the characters is too small to read. Could you enlarge more?

Response:  According to your suggestions, the size of the Figures 6 and 7 have been modified. 

  1. The text after Fig. 6 explained the simulation results of particle distribution in the box. However, I couldn’t catch the information on how these distributions influence methane detection.

Response: The influence of dust concentration on the detection accuracy of methane is shown in the measurement data in Table 4.

Question: Please add your explanation to the manuscript. Table 4 seems to be the list of dust concentrations.

Response: The influence of dust concentration on the detection accuracy of methane is shown in the measurement data in Table 5. The influence of dust concentration on the detection accuracy of methane is explained in the lines 337-343. I'm very sorry that the first reply was wrongly written as Table 4.

  1. Page 12, line 1-2; you wrote “High-humidity N2 was added to the chamber after pure water, in order to change the humidity of the simulated chamber, which was then passed into the TDLAS methane detection device.”. I think that experiments should be examined in the air atmosphere.

Response: Your suggestion is correct, the actual methane concentration measurement should be done in the air atmosphere. However, in order to obtain the influence of various environmental humidity on methane measurement, this paper designs a simulated experimental cabin to obtain the measurement of methane concentration under different humidity conditions.

Question : Your answer should be contained in the manuscript.

Response:  According to your suggestions, the explanation of the question is added in lines 346-351 of the manuscript.

   Thank you again for your comments, and we hope our responses and corrections would meet with your approval.

Reviewer 2 Report

I accept the responses to my comments and I am not submitting any more comments. I propose to qualify the manuscript for further stages of evaluation.

Author Response

      Thank you very much for your letter and for the reviewers’ comments concerning our manuscript entitled “Research on Methane Measurement and Interference Factors in Coal Mines” (ID: sensors-1790410). Those comments are all valuable and very helpful for revising and improving our paper, as well as the important guiding significance to our researches.

      Thank you again for your comments.